# Breast Milk Donation After Perinatal Loss: A Qualitative Exploration of Maternal Grief and Healing Among Israeli Arab Women and the Islamic Legal-Ethical Perspectives: A Qualitative Research Study

**DOI:** 10.3390/healthcare13243309

**Published:** 2025-12-17

**Authors:** Mahdi Tarabeih, Orsan Yahya, Mohammad Sabbah, Khaled Awawdi

**Affiliations:** 1School of Nursing Sciences, The Academic College of Tel-Aviv-Yaffa, Rabenu Yeruham Street, P.O. Box 8401, Yaffo 6818211, Israel; 2The Azrieli Faculty of Medicine, Bar-Ilan University, 8 Henrietta Szold St., Safed 1311502, Israel; orsanya@clalit.org.il; 3Department of Family Health, Clalit Health Service, Afula 1812201, Israel; 4Rambam Hospital Health Care Campus, 8 HaAliyah HaShniya St., Haifa 3109601, Israel; 5Department of Nursing, Faculty of Health Sciences, Ramat Gan Academic College, 87 Pinhas Rotenberg Street, Ramat Gan 5227500, Israel

**Keywords:** breast milk donation, perinatal loss, maternal grief, Islamic perspectives, milk kinship (*rida’a*), bereavement care, lactation, Interpretative Phenomenological Analysis (IPA)

## Abstract

**Highlights:**

**What are the main findings?**
Breast milk donation after perinatal loss served as a meaningful coping mechanism for bereaved Muslim mothers, helping them manage grief and maintain a symbolic bond with their lost child.Interviews with participating religious scholars indicated a consensus on the permissibility of milk donation when *rida’a* regulations are followed, suggesting that Islamic jurisprudence can provide supportive guidance for bereaved mothers in healthcare settings.

**What is the implication of the main finding?**
Healthcare providers should integrate structured counseling that includes milk donation options to support grieving mothers emotionally and psychologically, while being sensitive to cultural and religious considerations. Milk banks and medical institutions must implement practices aligned with Islamic ethical guidelines to respect religious concerns and provide compassionate, dignified care.

**Abstract:**

**Background/Objectives**: After perinatal loss, namely stillbirth and neonatal death, many bereaved mothers continue to produce breast milk, facing the decision as to whether to suppress lactation or donate their milk. Our aims were to explore the experiences and views of Muslim mothers who had donated their breast milk following perinatal loss and examine the Islamic legal-ethical perspectives relating to milk donation. This research explores how milk donation serves as a coping mechanism and how Islamic teachings frame its permissibility and ethical considerations. **Methods**: A qualitative research methodology was employed, using a Interpretative Phenomenological Analysis (IPA). Nine bereaved Muslim mothers who had donated their breast milk and three Islamic religious scholars (an Imam, a Mufti, and a Muslim jurist) participated in in-depth interviews. Thematic analysis identified recurring patterns and insights. **Results**: Our findings revealed that mothers experienced milk donation as a coping mechanism, allowing them to maintain a symbolic connection with their lost child while contributing to other infants’ survival. Religious scholars who we interviewed agreed that milk donation is permissible in Islam, provided that milk kinship (*rida’a*) regulations are observed. Mothers reported a strong need for structured support from healthcare providers and religious leaders in order to assist in the informed decision-making process. **Conclusions**: Breast milk donation after perinatal loss aids in grief management for bereaved mothers while benefiting vulnerable infants. Healthcare providers should offer comprehensive lactation counseling for bereaved mothers, including milk donation options. Milk banks should implement processes in alignment with *rida’a* guidelines. Improving support systems for bereaved mothers can alleviate their grieving process while ensuring alignment with cultural and religious norms.

## 1. Introduction

Perinatal loss encompasses a profoundly intricate grieving process related to miscarriage (pregnancy termination before 20 weeks), stillbirth (fetal death after 20 weeks), and neonatal death (death within 28 days) [1]. In addition to severe emotional and psychological distress, mothers may experience painful lactation. They must then decide whether to suppress lactation or donate milk. Donation has been recognized as therapeutic, helping bereaved mothers navigate grief and reconstruct maternal identity [2,3,4,5,6], yet it remains underexplored and inconsistently implemented across healthcare and cultural settings [7].

### 1.1. Perinatal Mortality

Stillbirth and neonatal death produce unique grief, as mothers mourn both their infant and the loss of their anticipated maternal role [8]. Emotions such as anger, guilt, anxiety, and depression are common [9], and the mother may develop PTSD, a low mood, or sleep disturbances [10]. Notably, prolonged complex grief occurs in ~20–30% of mothers. Broader social responses, such as silence, poor communication, or awkwardness can intensify the sense of loss [11]. Engagement with one’s emotions aids recovery, while meaning making predicts capacity to manage grief and achieve healing [12]. Some mothers even report post-traumatic growth, experiencing personal transformation after their loss [13,14].

### 1.2. The Benefits of Breast Milk

Breastfeeding provides numerous advantages, including optimal nutrition, gastrointestinal and psychological benefits, and bolstered immune protection [15]. Human milk contains antioxidative and immune-supportive properties [16], thus, lowering the risk of infectious diseases [17]. The World Health Organization (2015) [18] endorsed breastfeeding as the optimal nutrition for infants, and accordingly, NICUs increasingly rely on donor milk when mothers’ milk is unavailable [19]. Prescription of donor milk has steadily grown [11], with evidence linking it to improved neurodevelopmental outcomes in vulnerable, ill and preterm infants [20].

### 1.3. Bereaved Mothers and Their Breast Milk

After loss, lactation continues, requiring management. Options include suppression, gradual reduction, self-expression, or donation [21]. It is known that breast milk holds deep emotional significance for mothers, particularly after perinatal death [2]. In a study conducted in Australia, Welborn [3] found that expressing and donating milk helped bereaved mothers fulfill maternal instincts and process grief. For some, lactation affirms motherhood and memorializes the infant [22,23]. Milk can symbolize hope for mothers who nursed a sick infant or may become a poignant reminder of their loss. For infants who were stillborn, expressing milk may provide a way to navigate grief and maintain a connection with the deceased child [3].

### 1.4. Islamic Perspectives on Milk Donation

Milk donation involves religious-ethical concerns, particularly, the doctrine of milk kinship (*rida’a*), which creates familial bonds through breastfeeding [24,25]. This may subsequently complicate future marital prospects among individuals nursed by the same woman. Thus, the use of milk banks, where pooling obscures donor identity, presents difficulties in tracing kinship ties. Some scholars, i.e., Shaykh Ibn Saalih al-‘Uthaymeen, prohibited milk banks for this reason [24,25], while others, i.e., Yūsuf al-Qaradāwī, allowed them, citing that no religious objections exist [24]. Moreover, recent Islamic religious rulings show increasing support for milk donation within Islamic frameworks.

In 2023, the Minnesota Islamic Council issued a fatwa endorsing pasteurized donor milk, noting health benefits and minimal kinship risk [26,27]. In the UK, the United Kingdom Association for Milk Banking and the Muslim Council of Britain likewise supported donor milk, recommending robust systems to ensure traceability in order to mitigate *rida’a* concerns [27,28]. Though explicit rulings on post-loss milk donation specifically after fetal loss are rare, Islamic principles emphasizing the importance of preserving life and altruism support its permissibility [29].

### 1.5. Rationale for Study

Maternal grief spans a range of challenging emotions [30]. Our aims were to examine mothers’ experiences of expressing and donating breast milk during bereavement and examine the permissibility of this practice within Islamic teachings. Although, perinatal loss and breast milk donation have been examined in several cultural contexts, a clear gap remains in the literature. Firstly, very few studies have specifically focused on bereaved Muslim mothers, despite the unique religious, social, and ethical factors shaping their experiences. Secondly, no existing research has combined an Interpretative Phenomenological Analysis (IPA) with an Islamic legal-ethical analysis, leaving unexplored how religious jurisprudence interacts with mothers’ lived experiences. Thirdly, milk donation after perinatal loss remains clinically unsupported and ethically under-addressed in Muslim-majority contexts, where *rida’a*-related concerns and limited institutional guidance create significant ambiguity.

This study directly addresses these gaps by providing an in-depth phenomenological exploration of Muslim mothers’ experiences alongside contextual Islamic legal perspectives, offering novel insights for clinical, ethical, and religious guidance. It may contribute to healthcare by addressing the intersection of bereavement care, lactation management, and cultural-religious considerations, thereby offering guidance for clinicians working with Muslim families after a perinatal loss.

## 2. Research Methodology

### 2.1. Participants

The nine participating mothers, ranging in age from 24 to 41 years (mean = 31.5), all lived with a partner. Six were multiparous and three were primiparous. Educational backgrounds ranged from secondary education to university degrees or advanced professional training. Socioeconomic status varied, with participants representing both rural and urban communities. At the time of the interview, the interval since the mother’s loss ranged from three months to two years. Three had lost their firstborn immediately after birth, three experienced intrauterine fetal demise at 36–39 weeks, and three lost infants who were admitted to the NICU before passing away, indicating that some of the mothers had expressed breast milk before their loss. Sociodemographic data included parity, marital status, and family constellation, allowing us to explore how experiences differed across primiparous and multiparous mothers, and to include references to fathers and extended family. Participants came from diverse socioeconomic backgrounds, including both urban and rural communities. All women reported being emotionally stable, enough to participate, based on a pre-interview well-being screening.

In addition, three Islamic scholars (an Imam, a Mufti, and a jurist) were interviewed; they ranged in age from 51 to 64 and had advanced training in Islamic law.

### 2.2. Inclusion and Exclusion Criteria

The study included Sunni Muslim women in Israel who experienced perinatal loss (stillbirth or neonatal death) and who chose to donate their breast milk to Israel’s human milk bank. Exclusion criteria were women who did not experience perinatal loss, who did not donate breast milk, or who were not Muslim. Exclusion criteria were women with ongoing severe psychiatric illness, inability to provide informed consent, or insufficient Hebrew or Arabic proficiency to participate in an in-depth interview

### 2.3. Recruitment Process

The first author, a practicing Muslim, personally contacted nine women and three religious leaders: an Imam, a Mufti, and a Muslim jurist. The religious leaders were invited to participate in the study in order to explore their experiences and perspectives regarding breast milk donation following fetal death. Each religious leader represented a different community in the northern, central, and southern regions of Israel. All participants provided both verbal and written informed consent before participating.

### 2.4. Research Design

The Interpretative Phenomenological Analysis (IPA) was chosen as the research methodology allowing for a structured and in-depth analysis of lived experiences, emphasizing both phenomenological meaning-making and interpretative contextualization within health-related frameworks. Creswell [31] contends that IPA emphasizes deep exploration of “lived experiences” over broad participant samples, making it ideal for studying breast milk donation after pregnancy loss—a complex and sensitive topic best understood through rich qualitative data [32]. IPA employs “double hermeneutics,” combining interpretative analysis—where researchers contextualize participant data within psychological frameworks—and phenomenological analysis [33], rooted in Heidegger and Gadamer’s philosophies [34], to explore and give voice to participants’ lived experiences and personal meanings. This idiographic focus allows for a systematic, in-depth understanding of individual experiences.

#### 2.4.1. IPA Analytic Process and Coding Stages

All transcripts were read and re-read multiple times by the first and second authors to enable immersion in participants’ narratives. This iterative reading allowed the researchers to bracket initial assumptions, attend to emotional tone and linguistic expression, and develop an intimate familiarity with each participant’s lived experience.

Reflexivity was integral throughout data collection and analysis. Given the first author’s identity as a Muslim clinician, reflexive journaling and bracketing as well as analytic decisions were discussed within the research team to ensure interpretive balance without suspicion of power or relational bias. Participants were repeatedly informed that participation was voluntary, confidentiality was guaranteed, and declining or withdrawing would have no consequences. To mitigate the influence of personal assumptions and cultural familiarity, regular analytic meetings enabled critical dialogue regarding potential biases. A standardized interview protocol and adherence to IPA’s idiographic commitment helped maintain analytic rigor. These measures in aggregate ensured that interpretations remained grounded in participants’ narratives rather than the researchers’ preconceptions.

Exploratory comments were generated through line-by-line analysis focusing on descriptive content, linguistic features (e.g., metaphors, emphasis, affective tone), and conceptual reflections. These exploratory notes served as the analytic bridge between raw narrative and emergent themes. The five emergent themes were developed by synthesizing clusters of exploratory notes within each transcript. This interpretative step involved condensing meaning units, identifying salient psychological processes, and articulating the central experiential patterns expressed by each participant.

Connections between emergent themes were examined through abstraction, subsumption, and contextualization. The research team developed a thematic map for each participant, identifying conceptual relationships, convergence, and divergence. Related themes were grouped into superordinate categories, which later informed the cross-case analysis. Cross-case comparisons were conducted after completing idiographic analyses for all participants. The research team compared thematic structures across the twelve interviews (nine mothers and three religious scholars), identifying patterns of convergence, points of divergence, and the shared experiential structures that formed the eight final themes. Themes were included only if they were present in multiple accounts or held strong conceptual relevance.

#### 2.4.2. Two-Layer Interview Design

The study employed a two-layer qualitative design, reflecting the different epistemological orientations of the participant groups.

Layer 1 consisted of phenomenological interviews with bereaved Muslim mothers, analyzed using Interpretative Phenomenological Analysis (IPA) to capture lived experience, emotional meaning-making, and the embodied process of breast milk donation after perinatal loss.

Layer 2 consisted of key-informant interpretive interviews with Islamic religious scholars (an Imam, a Mufti, and a jurist). These interviews served a contextual and jurisprudential function, providing interpretive insights regarding *rida’a*, milk kinship, and ethical permissibility in Islam. Whereas the mothers’ interviews formed the idiographic experiential foundation of the study, scholars’ interviews provided an analytic contextual layer that illuminated the religious-ethical environment in which mothers’ decisions were embedded. These two layers were analyzed separately and later integrated to enrich the interpretation of findings.

### 2.5. Sample Size and Data Saturation

The sample size was guided by the principle of *data saturation* [35]; thereby, data collection continued until recurring insights and a stable representation of the shared reality emerged. Data adequacy was achieved when no new experiential meanings were emerging, consistent with IPA’s emphasis on depth rather than breadth. Following multiple iterative readings of the transcripts and the development of emergent and superordinate themes, the research team determined that additional interviews were no longer generating novel experiential insights. This point was, therefore, considered sufficient to establish analytic adequacy within the IPA framework

### 2.6. Materials

The first author initiated telephone contact with the participants explaining to them the research objectives and scope. Subsequently, an informed consent form, essential for ensuring ethical compliance and participant understanding was distributed to the participants via the WhatsApp application. The participants were asked to sign and return the form. Upon receipt of the signed consent forms, the first author proceeded to organize individual in-depth interviews. Interviews with women were conducted at the Mother and Child Clinic of their choice, whereas interviews with religious leaders were held in face-to-face meetings at a mosque and at the Islamic Fatwa Center, thereby creating a contextually relevant and comfortable environment for each participant in accordance with the research framework.

### 2.7. Ethical Considerations

Ethical approval for this study was obtained from the Ethics Committee of the academic institution affiliated with the second author, approval number 2023-1010-MTA. Participants were required to sign an informed consent form detailing the terms of their participation and the conditions of publication of the study’s findings. To maintain confidentiality and privacy, demographic data were aggregated and presented exclusively at the group level, following the recommendations of Morse [36]. The informed consent document stated that participation was both anonymous and confidential and assured participants of their right to withdraw from the interview at any time. Furthermore, the document informed participants that selected excerpts from their interviews might be included in future publications, thereby ensuring transparency in the research process.

### 2.8. Interview Protocol

The interview began with a formal greeting, accompanied by well wishes for the participants’ health. The second author expressed appreciation for their participation, followed by a detailed reading of the informed consent form which included a clear explanation of the research objectives, methodology, assurance of the participants’ autonomy to terminate the interview, assurances of confidentiality and anonymity, and a request for consent to record the interviews. Subsequently, participants were asked to confirm their willingness to continue their participation in the study.

Following the methodological framework proposed by Josselson [37], the interview was structured around a single, comprehensive question, allowing participants the flexibility to respond as they saw fit. The session began with an informal dialogue to establish rapport and set the participant at ease. The duration of the interview was generally between 60 and 70 min, following the guidelines suggested by Roberts [38]. The primary question posed during the interview was: “**Please articulate your views and the Islamic perspective regarding milk donation.**”

This question functioned as the initial gateway to a deeper phenomenological exploration, employing a series of structured probing strategies to elicit rich, first-person accounts of the mothers’ embodied, emotional, and meaning-making experiences. Probes included experiential follow-ups (e.g., “Can you describe what that moment felt like in your body?”, “What thoughts or emotions accompanied that experience?”), temporal sequencing questions (“What happened next?”, “How did this change over time?”), and reflective or interpretive prompts (“What did this mean for you as a mother?”, “How did this affect your sense of connection to your baby?”). For mothers, additional prompts examined motivations, emotional ambivalence, social reactions, and spiritual interpretations. Interviews with religious scholars incorporated conceptual probes to clarify jurisprudential reasoning, ethical concerns, and interpretive variation.

Throughout the interview, a friendly and non-judgmental atmosphere was consciously maintained, both in verbal and non-verbal communication. The interviews were meticulously recorded by the second author and subsequently transcribed. To preserve participant anonymity, each individual was assigned a unique identification code prior to transcription.

### 2.9. Interviews

#### 2.9.1. Participant Readiness and Emotional Well-Being at Time of Interview

Interviews were conducted between March and November 2024, ranging from three months to two years after the loss, depending on each mother’s readiness to participate. This time frame aimed to balance the immediacy of grief with participants’ emotional capacity to engage meaningfully in research. Prior to each interview, a brief well-being and mental health screening ensured participants were emotionally stable enough to reflect on their experiences. Women in acute distress were referred to psychosocial services and only interviewed when comfortable. A distress protocol was in place, and follow-up support was offered. Additionally, a follow-up phone call was offered one week after the interview as a means of consultation and support, if needed. Participation in the follow-up call was optional. Many participants found meaning and comfort in sharing their stories, often linking milk donation to their coping process.

#### 2.9.2. Interview Topics and Key Findings

The interviews covered various aspects, including milk production, motivations for milk donation, Islamic perspectives, religious leaders’ opinions, the husband’s perspective on milk donation, the role of the milk bank, staff support, and the connection to maternal milk, and decision-making processes. Participants were encouraged to provide detailed accounts of their experiences and share how they found meaning in their journey. As each discovery during the interview was a product of the relationship between the participants and the researcher, the emphasis on demonstrating empathy and sincerity remained a central methodological component [39].

#### 2.9.3. Key Themes Identified from the Interviews

The findings from our interviews were categorized into five key themes:Breast milk donation after pregnancy loss; a meaningful practice based on Islamic values.Sustaining a bond and creating a lasting legacy for the baby.Personal healing through donation: navigating pain, coping, and emotional restoration.A sense of mission and solidarity contributing to infant survival and the wider community.Negotiating social responses within a supportive donor environment.

## 3. Results

### 3.1. Theme 1. Breast Milk Donation After Pregnancy Loss as a Meaningful and Value-Congruent Practice

Women who experienced perinatal loss frequently described breast milk donation as a deeply meaningful act aligning with their moral and spiritual values. Rather than abruptly suppressing lactation—a physically and emotionally painful process—donation offered a purposeful alternative [40,41,42]. Providing milk for vulnerable infants imbued their loss with significance, helping them navigate the overwhelming sense of emptiness that followed the death of their baby, restoring their sense of agency and transforming their pain into an act of altruistic giving and healing [43].

Testimonies from donating mothers illustrate the emotional experience associated with this act. Hanan, who experienced a stillbirth: “*Knowing that my breast milk could save lives provided me with a purpose during a time when I felt as if I had lost everything.*” Layla, who lost her baby in the 8th month of pregnancy: “*Donating was my way of giving life, providing fulfillment and comfort, even after losing my baby*.” Khadijah, whose infant died: “*The donation allowed me to transform my loss into a positive act, offering another baby what mine never had the chance to receive*.” Asmaa, who lost her daughter during child-birth: “*Knowing that my breast milk was helping other babies was a ray of light in the darkest period of my life*.” Fatima, speaking as a physician and as a mother who experienced pregnancy loss, shared: “*In medicine, we talk about evidence-based healing, however, grief does not work that way. Donating my milk was the only real thing I could hold onto, it helped me heal*.” (Appendix A).

This meaning-making to regain a sense of agency naturally leads into Theme 2, in which milk donation becomes not only an altruistic act but also a deeply relational one. Women describe the donation as a continuation of mothering, a way to maintain a bond with their baby, and to create a lasting legacy in their child’s name.

### 3.2. Theme 2. Sustaining a Bond and Creating a Lasting Legacy for the Baby

Many participants described milk expression and donation as a continuation of the maternal role that had been abruptly interrupted by loss. Breast milk donation allowed women to feel that they were still providing nourishment, and that their baby continued to have an impact, even indirectly, on the lives of others. Across testimonies, women emphasized that donation represented their “last way” of mothering.

Hanan reported:


*“Every drop of milk I managed to donate felt like a love letter to my baby. He is not here with me, but because of him, other babies will receive life and health which gave me the strength to keep on going.”*


Layla speaks of the connection to her lost son that donating her milk represented:


*“At first, I didn’t want to pump at all. It constantly reminded me of my pain. But once I started donating, I felt as if my son was still present in some way—he was giving to others through my milk.”*


Khadijah spoke of remembering her daughter through her milk donation:


*“My heart broke when I lost my daughter, but knowing that my breast milk helped save other babies gave me the strength to continue. I believe that Allah gave me the opportunity to do something meaningful in her memory.”*


Umm Salama, who experienced a stillbirth at six months, recounted:


*“When I started donating, I felt as if I was holding my baby—just in a different way. I knew that this act of giving would bring blessings to both of us, and that perhaps one day, in the afterlife, I will meet him and tell him how he helped others.”*


Soraya: *“I wasn’t ready to say goodbye. When I started donating, it felt like I was still part of the world of motherhood, even though I had lost my baby. Every pumping session was a moment of connection.”*

Asmaa related that she felt helping other children was a way of maintaining her own infant’s presence in the world.

*“I thought everything had ended when she passed away, but then I realized that my milk is not just a memory of her, but also a gift to other children. Through every donation, she remains part of this world.”* (Appendix B).

This ongoing sense of connection naturally transitions into Theme 3, where milk donation not only maintains the maternal bond but also becomes a source of emotional restoration.

### 3.3. Theme 3. Healing Through Donation: Navigating Pain, Coping, and Emotional Restoration

The process of pumping and donating milk was described as a complex emotional experience in which grief, physical pain, hope, and fulfillment coexisted. For many, the act of breast milk donation provides women with the opportunity to channel their pain into a positive act, thus, transforming raw grief and a sense of helplessness into restorative purpose and creating a meaningful healing process connecting the body, mind, and faith [44,45]. Soraya, a mother whose infant daughter died in the NICU:


*“During the first few days after losing my daughter, I could not find the strength to get out of bed, but when I started pumping and donating, it gave me a sense of purpose, instead of feeling powerless, I felt like I was doing something good. Every pumping session was painful, but every donation brought satisfaction. Donating milk helped me not only emotionally, but also strengthened my faith that there was a purpose to my pain.”*


Amira, a lawyer whose infant was born prematurely, said:

*“The pain didn’t go away, but at least I found a way to channel it into something positive. Instead of being consumed by my grief, I felt that I could turn my loss into an act of giving.”* Asmaa also mentions this sense of purpose: *“I felt helpless, but providing my milk to a baby in need, gave me back a sense of purpose.”*

Umm Salama added:


*When I held the full bottle of milk, I felt as if I was still giving something of myself to my child. That brought me comfort amidst my pain.*


Umm Salama: *“The pain became part of something greater.”*

Layla: *“This was the only way I could turn my loss into something meaningful.”*

Khadijah also said: *“My pain didn’t disappear, but it turned into something meaningful.”*

Amira said: *“Knowing my milk helped another baby gave me strength.”*

Fatima spoke of growing from the tragedy: *“I chose to take my pain and turn it into something beneficial.” Donating was my way of finding light in the darkness.”*

Khadijah assigned an additional meaning: *“This was my farewell process.”*

Amal: *“Pumping allowed me to say goodbye at my own pace.”*

Fatima expressed her feelings that donating her breast milk was her personal decision taken freely and restoring her sense of agency: “*I could have stopped immediately, or I could have taken my pain and turned it into something beneficial for others. I chose the latter.*”

Theme 3 demonstrates how breast milk donation becomes a vital pathway of emotional restoration for bereaved mothers. The women felt that their pain contributed to something greater and helped them move through grief with intention and dignity and a sense of purpose (Appendix C).

This internal process of healing leads to Theme 4, where the personal transformation experienced through donation expands outward into a sense of mission and communal solidarity. Donating milk shifts from being solely a coping strategy to becoming a purposeful contribution to infant survival and a shared bond with other mothers.

### 3.4. Theme 4. Mission and Solidarity: Contributing to Infant Survival and the Wider Community

For many women who lose infants, breast milk donation becomes a mission in knowing that they are providing life to sick and premature infants, as well as children who lack access to their own mother’s milk. This strengthened their sense of solidarity, and established a support network founded on compassion, hope, and generosity, and has been demonstrated in studies conducted across regions with a predominately Muslim culture, such as Nigeria [46] and Yemen [47].

Hanan, who donated milk after experiencing a stillbirth, related: “*Instead of giving only to my baby, I was giving life to many others.*”

Asmaa spoke of her sense of connection to the community: “*I never met the mothers, but I felt connected to their children in a unique way. A part of me is with their children, and that felt like a divine blessing.*”

Khadijah: *“My milk helped save others. That gave me hope.”*

Soroya: *“I felt that I was doing something good—for the babies and for myself.”*

Khadijah: *“My pain didn’t disappear, but it turned into something meaningful by helping others.”* (Appendix D).

Through donating, women transformed their grief into collective care, positioning themselves as part of a wider community committed to nurturing vulnerable children (Appendix D).

This expanding sense of mission brings the discourse to Theme 5, where the women’s engagement with community and solidarity intersects with the social responses to their donation. Theme 5 shows how supportive donor environments, such as milk banks, healthcare workers, friends, and informal networks became essential in buffering these negative reactions, validating the mothers’ choices, and sustaining the communal ethos highlighted in Theme 4.

### 3.5. Theme 5. Negotiating Social Responses Within a Supportive Donor Environment

Women navigated varied social reactions to their decision to continue pumping after their loss. Some encountered misunderstanding, discouragement, or social pressure to stop lactation immediately. These experiences contrasted with the highly supportive environment provided by milk banks, healthcare workers, and donor communities, who validated the women’s choices and treated them with sensitivity.

At times they faced discouraging responses to their decision to donate such as: “*Why are you still pumping?”; “This will only cause more pain;” “You should stop and face the loss.*”

Soraya recalls a similar experience: “*People around me didn’t understand why I was doing this. They thought I should stop and move on.*”

Khadijah shed light on what motivated her to ignore such comments.


*“Some asked why I didn’t just stop pumping. But for me, this was a way to gradually say goodbye, at my own pace.”*


Amira justified her decision, saying: “*I knew that society wouldn’t always understand, but I chose my own path. And Allah sees the intentions in my heart.*”

In many Muslim communities, women support one another during times of hardship through informal networks that provide a sense of belonging and understanding to mothers who have experienced loss.

Umm Salama: *“It was a circle of support from which I drew strength.”*

Hanan: *“My friends never let me feel alone.”*

Human milk banks, medical institutions, and community organizations may also play a crucial role in supporting these mothers.

Layla decided to donate at a milk bank. She reported:


*“The place where I donated was incredibly supportive and welcoming. They didn’t just collect my milk, they understood my pain, spoke to me with respect, and made me feel that my donation was valuable. It helped me feel as if I wasn’t alone”.*


Asmaa describes her experience:


*“I arrived at the milk bank with a lot of fears and mixed emotions. But the staff was so compassionate, they respected what I had been through, supported me, and didn’t make me feel like I was just another donor. They gave me space to process my grief.”*


Fatima spoke in a similar vein:


*“What helped me the most was that the milk bank staff understood my pain. They didn’t just take my milk; they spoke to me with kindness and thanked me for my donation, and that made an enormous difference.”*


### 3.6. Practical Considerations

Breast milk donation is a highly commendable act in Islam; however, recipient infants must be properly documented in order to prevent future legal complications regarding marriage. Because of the *rida’a* guidelines, this carries significant implications, particularly, concerning marriage restrictions, and thus, societal repercussions [48]. According to Islamic jurisprudence, if a baby consumes breast milk at least five full times from a woman who is not their biological mother, they are considered milk siblings (*rida’a*), making them legally related to the donor mother’s biological children. Therefore, careful documentation of milk donations and recipient identities are essential, especially when donations are facilitated through milk banks. Documentation practices were valued for clarity in managing milk-kinship implications, though women emphasized that support, empathy, and respectful treatment mattered even more.

## 4. Discussion

For many professional mothers, despite their structured careers and personal achievements, the death of a child often disrupts their sense of identity. The bereaved mothers who were interviewed frequently described donating milk as a meaningful and purposeful response to grief, one that restores a sense of agency and allows continued contribution, both medically and emotionally [49]. Milk donation provides a way for such women, especially those who are used to control and structure in their professional lives, to reclaim a sense of purpose after the abrupt loss of agency caused by stillbirth or neonatal death. Many professional mothers viewed milk donation as an extension of their core identity—the desire to contribute meaningfully to society. The feeling of maintaining a bond with their lost child by helping to nourish other infants offered not only solace but a renewed sense of identity.

The experiences of Muslim mothers who donated breast milk after perinatal loss align with cross-cultural research on parental bereavement. Across Western, African, and Asian contexts, mothers commonly seek to maintain an emotional bond with the deceased child through symbolic or altruistic acts [50,51]. In our study, breast milk donation functioned as such an act. The Muslim mothers’ narratives echo findings from Nigeria, Australia, India, and Europe, where parents oscillate between pain and fulfillment as they affirm relational continuity while dealing with post-loss emotional instability [52,53].

Within this broader cross-cultural landscape, Islamic perspectives provided a distinctive interpretative framework that mediated mothers’ decision-making and shaped emotional meaning. Concepts such as ‘qadar’ (divine decree), ‘ajr’ (spiritual reward), and ‘sadaqah jariyah’ (continuous charity) enabled many mothers to reinterpret perinatal loss as a morally valuable opportunity to help other infants. This theological reframing reduced decisional conflict and supported agency, particularly when religious scholars affirmed the permissibility of donation under regulated *rida’a* conditions. Mothers described how such guidance alleviated fear, guilt, or confusion, consistent with literature showing that Islamic jurisprudential reassurance plays a key role in reproductive ethics and coping [54,55].

A core dimension of the IPA concerned sense-making. Mothers used milk donation to impose coherence on a loss that otherwise felt senseless, articulating narratives in which their infants “continued to give life” or “remained present” through the milk. This mirrors global findings that embodied rituals can anchor parents during acute loss.

Social-relational processes strongly shaped these experiences. While some encountered skepticism or stigma from relatives or community members, many described networks of solidarity among other bereaved mothers, lactation staff, and religious leaders. Parity and family structure further influenced meaning: primiparous mothers tended to describe donation as a redefinition of first-time motherhood, whereas multiparous women framed it as continuity with an established maternal identity.

Across narratives, identity reconstruction emerged as a central theme. Loss had fractured maternal identity, whereas donation allowed mothers to reclaim this identity no longer in relation to their own infant, but through altruistic contribution to vulnerable infants. These processes converged into existential themes: mothers grappled with the spiritual meaning of suffering, questioned divine purpose, and ultimately found solace in the belief that their grief could generate life for others. The notion of a “legacy of love” emerged repeatedly, suggesting that for many mothers, milk donation reframed their child’s brief existence into a broader narrative of compassion, religious devotion, and communal benefit.

The integration of Islamic jurisprudential perspectives also illuminated how ethical, legal, and emotional dimensions interact in real-world decision-making. Mothers’ reliance on scholars to interpret *rida’a* regulations reflects how religious authority structures ethical comfort and mitigates the fear of violating marriage restrictions or moral boundaries. Our findings show that when religious rulings were framed in supportive, evidence-informed ways, mothers felt empowered; when rulings were restrictive or ambiguous, emotional distress intensified.

### 4.1. Clinical Implications

The findings carry significant implications for healthcare practice. First, culturally competent lactation counselling is essential. This means counselling that explicitly integrates religious values, acknowledges *rida’a* concerns, and affirms the legitimacy of ambivalent emotions. Evidence from culturally attuned maternal-health models shows that such alignment strengthens trust and improves psychological outcomes [56,57].

Second, milk banks serving Muslim populations require structured, transparent, and ethically grounded protocols that address documentation, traceability, and informed consent in ways that respect Islamic jurisprudential sensitivities. Emerging international models demonstrate that when milk-bank processes incorporate religious guidelines, community acceptance increases and conflict diminishes [58,59].

Third, coordinated training between religious leaders and healthcare teams is needed to ensure consistency in counselling. Mismatched messages where clinicians advocate donation but local imams express caution can destabilize mothers emotionally and impede coping. Joint educational initiatives have been shown to reduce conflict and support family well-being [60,61].

### 4.2. Recommendations for Further Study

While the study provides valuable qualitative insights, a larger sample size across different geographical regions and cultural contexts would strengthen the generalizability of the findings. Future research could include a more diverse group of Muslim mothers, religious scholars, and medical professionals from different countries. Although the study included three Islamic scholars, supplementing viewpoints from scholars across different Islamic schools of thought (Sunni, Shia, and other various sects), could provide a more comprehensive understanding of religious perspectives on milk donation after fetal loss.

Incorporating a quantitative component, such as standardized grief and psychological well-being assessments would allow for statistical comparisons between mothers who choose to donate milk and those who do not, offering a deeper insight into the psychological benefits of milk donation. For policy-making purposes, more in-depth discussion of the legal and ethical implications of breast milk donation in Islamic contexts, particularly regarding the documentation of *rida’a* would help policymakers craft appropriate guidelines. In particular, providing specific guidelines on how milk banks can operate within Islamic ethical frameworks (e.g., clear donor-recipient documentation to mitigate *rida’a* concerns), would render the research more practical for healthcare institutions [61,62].

### 4.3. Integration of Mothers’ Phenomenological Themes with Scholars’ Interpretive Perspectives

As the study employed a two-layer qualitative design, the integration of bereaved mothers’ experiential themes (Layer 1) with Islamic scholars’ interpretive perspectives (Layer 2) is essential for understanding the full context of breast milk donation after perinatal loss. While IPA requires idiographic presentation of mothers’ lived experiences, the interpretive insights provided by religious scholars offer a complementary framework that shapes, informs, and sometimes mediates mothers’ meaning-making processes.

The alignment between maternal narratives and Islamic principles suggests that religious teachings may provide moral validation, spiritual meaning, and emotional reassurance for bereaved women navigating complex decisions about milk donation.

At the same time, certain divergences emerge. Mothers focused primarily on personal, emotional, and embodied experiences—grief, purpose, and maternal identity—whereas scholars emphasized jurisprudential concerns related to *rida’a* (milk kinship), documentation, and lineage preservation. Their intersection, however, becomes particularly meaningful when mothers seek religious endorsement or face uncertainty regarding permissibility.

Integrating both layers reveals a holistic understanding: mothers’ experiential desire to heal through donation operates within an Islamic ethical ecosystem that simultaneously encourages altruism and mandates clear documentation to avoid lineage complications. This integration enhances clinical applicability by illustrating how healthcare providers can support bereaved Muslim mothers through emotionally sensitive counseling that acknowledges both their lived experiences and the religious-ethical frameworks that inform their decisions. Ultimately, the synthesis of both layers enriches the understanding of milk donation after perinatal loss as not merely an individual coping mechanism but as an experience situated at the intersection of personal grief, religious meaning, ethical guidelines, and community norms.

### 4.4. Limitations

Our study did not include the father’s perspectives on his wife’s donation of breast milk. We focused on the mothers’ experiences but did not consider how fathers or other family members influence the decision to donate their wife’s breast milk. In many Islamic societies, family decisions, particularly, those related to religious rulings, often involve husbands, extended family, or religious authorities.

The study lacked longitudinal data, while capturing the short-term experiences of the mothers who donated the milk, it did not track the long-term psychological, emotional, or religious implications of their decision. This made it difficult to assess its lasting impact on grief resolution. A follow-up study could examine whether milk donation has lasting therapeutic effects or leads to future regret or reconsideration.

## 5. Conclusions

This study highlights the increasing need to recognize breastfeeding following perinatal loss as a significant factor in a mother’s emotional coping process. The expression of breast milk played a central, powerful, and positive role for all participants in this study. Mothers reported a strong need for broad support from the Muslim community, religious scholars, and healthcare professionals in order to obtain the necessary knowledge and guidance for making informed decisions following perinatal loss.

Among the women who participated in the study, the process of expressing and donating breast milk functioned as a healing ritual, allowing them to partially fulfill their maternal role. Through this act, they experienced a sense of purpose and were able to associate positive meanings with their loss. The time and effort invested in commemorating their infant in this manner were essential to their emotional processing.

Breast milk donation after fetal loss represents a deeply personal, emotional, and religiously significant act. For the mothers, it offered a structured means to cope with their grief, maintain a connection with their lost child, and contribute meaningfully to society. Islamic scholars overwhelmingly support milk donation as an act of charity and healing, as long as it adheres to their ethical guidelines. Ultimately, breast milk donation embodies a powerful intersection between personal healing, religious faith, and social contribution, hence, allowing bereaved mothers to find meaning in their loss while offering life to others.

## Data Availability

Individual level data cannot be made publicly available due to legal and ethical restrictions. Aggregative data might be provided upon reasonable request to the corresponding author.

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
