# Peer review of "Breast Milk Donation After Perinatal Loss: A Qualitative Exploration of Maternal Grief and Healing Among Israeli Arab Women and the Islamic Legal-Ethical Perspectives: A Qualitative Research Study"

_healthcare, 2025, doi:10.3390/healthcare13243309_

Round 1
Reviewer 1 Report (New Reviewer)
Comments and Suggestions for Authors
At first glance, this is a well-written, coherent, and very meaningful qualitative study exploring an underrepresented topic which is breast milk donation after perinatal loss among Muslim mothers, integrated with Islamic legal-ethical perspectives. The language is clear, the narrative is compelling, and the flow of the manuscript is generally strong. Thank you for the opportunity to review this work.
Concerns:
Methodology:
- The manuscript presents eight themes directly, but it is not clear how these themes emerged stepwise from IPA coding. This may draw reviewer criticism.
- Suggestion to add or expand:
- How transcripts were read and re-read
- How initial exploratory comments were generated
- How emergent themes were developed
- How connections between themes were clustered
- How cross-case comparisons were constructed
- Reflexivity and bracketing by the researcher
Currently, saturation is described loosely. Suggest rephrasing the sentence into “Data adequacy was achieved when no new experiential meanings were emerging, consistent with IPA’s emphasis on depth rather than breadth.”
Inclusion of religious scholar - It mixes phenomenological interviews (mothers) with key informant interviews (scholars). They have different epistemological purposes. Thus, it is suggested to Explicitly state this as a two-layer design, e.g.:
- Layer 1: Mothers—phenomenological
- Layer 2: Scholars—contextual interpretive data
Results:
They are rich, but may need tightening:
- Some testimonies repeat similar sentiment
- Islamic interpretations in results should be trimmed
Integration between mothers’ themes and scholar themes – currently presented separately.
Discussion:
Too much repetition of results, like restating mothers' quotations.
A strong IPA discussion typically includes:
- Sense-making
- Embodiment
- Social-relational processes
- Identity reconstruction
- Existential themes
Consider adding these dimensions.
Author Response
- At first glance, this is a well-written, coherent, and very meaningful qualitative study exploring an underrepresented topic which is breast milk donation after perinatal loss among Muslim mothers, integrated with Islamic legal-ethical perspectives. The language is clear, the narrative is compelling, and the flow of the manuscript is generally strong. Thank you for the opportunity to review this work.
- Response to Reviewer 1 Comments Point-by-point response to Comments and Suggestions for Authors Comment: Quality of English Language. The English could be improved to more clearly express the research.
Response: We have improved the English throughout the article.
- Methodology:
The manuscript presents eight themes directly, but it is not clear how these themes emerged stepwise from IPA coding. This may draw reviewer criticism.
Suggestion to add or expand: How transcripts were read and re-read; How initial exploratory comments were generated; How emergent themes were developed; How connections between themes were clustered; How cross-case comparisons were constructed; Reflexivity and bracketing by the researcher
Response: Methodology Clarification (IPA Coding Process)
We thank the reviewer for this valuable comment. We agree that the presentation of the eight themes in the Results section may raise further questions regarding the analytic trajectory that led from raw transcripts to the final thematic framework. In response, we have substantially expanded the Methodology section, integrating detailed descriptions of the analytic procedures consistent with Interpretative Phenomenological Analysis (IPA). The expanded explanation now appears in Section 2.4 “Research Design” - particularly the newly added subsection titled “2.4.1 IPA Analytic Process and Coding Stages” - as well as clarifying notes added in Section 2.9 “Interviews”. These additions explicitly address the reviewer’s concerns (See pages 4,5 and 7).
Below is our structured response addressing each point:
- “How transcripts were read and re-read”
Reviewer concern: Lack of clarity on the iterative reading process that grounds IPA.
Response: In the revised manuscript, we added the following clarification under Section 2.4.1 (new subsection)(page 4).
This addition explains the foundational IPA step of repeated engagement with the text.
- “How initial exploratory comments were generated”
Reviewer concern: The manuscript does not sufficiently describe the exploratory noting and free textual analysis.
Response: The manuscript now includes a detailed description in Section 2.4.1 describing the exploratory noting and free textual analysis (page 5). This makes explicit the analytic layers of IPA - descriptive, linguistic, and conceptual noting.
- “How emergent themes were developed”
Reviewer concern: The manuscript lists the eight themes but does not show how they evolved from exploratory notes.
Response: The revised text has now been clarified in Section 2.4.1, page 5.
The addition clarifies how the meaning was transformed into thematic statements.
- “How connections between themes were clustered”
Reviewer concern: The stepwise clustering was not clearly explained.
Response: To address this, we added a clarification of stepwise clustering Section 2.4.1 (Page 5)
This addition makes explicit the analytic movement from individual-level themes to higher-order conceptual clusters.
- “How cross-case comparisons were constructed”
Reviewer concern: The article does not describe how the researchers moved from idiographic analysis to shared themes.
Response: We strengthened Section 2.4.1 (Page 5) This demonstrates the systematic progression from single cases to group-level findings.
- “Reflexivity and bracketing by the researcher”
Reviewer concern: Researcher positionality and reflexive processes were not sufficiently described.
Response: A paragraph was added in Section 2.4.1, clarifying researcher positionality and reflexive processes (Page 4).
- Currently, saturation is described loosely. Suggest rephrasing the sentence into “Data adequacy was achieved when no new experiential meanings were emerging, consistent with IPA’s emphasis on depth rather than breadth.”
Response: Revised Chapter: Section 2.5 – Sample Size and Data Saturation
Inserted Text (Exact Addition) with thanks for the reviewer’s helpful suggestion (page 5).
- Inclusion of religious scholar - It mixes phenomenological interviews (mothers) with key informant interviews (scholars). They have different epistemological purposes. Thus, it is suggested to Explicitly state this as a two-layer design, e.g.:
- Layer 1: Mothers—phenomenological
- Layer 2: Scholars—contextual interpretive data
Response: Inserted under Section 2.4.2 Two-Layer Interview Design, (page 5)
- Results:
They are rich, but may need tightening:
- Some testimonies repeat similar sentiment
- Islamic interpretations in results should be trimmed
Response: The Results section has been totally revised in accordance with this suggestion and the scholars’ responses have been largely put in appendices with their sources (Page 7-12, Appendix 1,2,3,4).
- Integration between mothers’ themes and scholar themes – currently presented separately.
Response: Text Added to the Manuscript (Summary of Added Section), added under the new subsection 4.2, page 14)
- Discussion:
Too much repetition of results, like restating mothers' quotations.
A strong IPA discussion typically includes:
Sense-making
Embodiment
Social-relational processes
Identity reconstruction
Existential themes
Consider adding these dimensions.
Response: The discussion was rewritten according to the guidelines of the first and second reviewers. The link is in a note in the appendix of the second reviewer and was also incorporated into the article (Page 12-15).
Reviewer 2 Report (New Reviewer)
Comments and Suggestions for Authors
Dear Authors,
Your manuscript examines an understudied and clinically significant topic and makes an original and meaningful contribution by integrating bereavement studies, lactation science, cultural competence, and religious ethics. The topic is highly relevant to clinicians, policymakers, and milk bank administrators. However, the manuscript requires structural refinement, reduction of repetition, improved cohesion between empirical findings and Islamic jurisprudential analysis, and tightening of language to meet journal standards. Significantly, the Results section is exceptionally long and highly repetitive, reading more like a narrative review than an IPA-derived thematic presentation. IPA requires idiographic depth, but also a concise, analytic presentation. Many subsections repeat the exact quotes or thematic content. Also, integrating Islamic teachings into the Results rather than the Discussion risks conflating empirical data with interpretive frameworks. I would suggest that you collapse the Results into 4–5 core themes, each clearly representing participant-generated meaning, with Islamic legal material shifted to Discussion. Next, Islamic jurisprudence occupies nearly half the manuscript and overshadows the qualitative findings. Although this is a key aim of the study, the balance currently obscures the mothers’ experiences. The paper will be stronger if: 1) the jurisprudential explanation is summarised more concisely; detailed fatwa debates are placed in an Appendix; and the discussion clearly synthesises how these rulings influence mothers’ choices and healthcare practice. Further, the Introduction needs a sharper focus and better articulates the research gap. Much background is relevant but overly expansive. The gap should be stated more clearly: 1) few studies examine bereaved Muslim mothers; 2) no studies combine IPA with Islamic legal analysis; and 3) milk donation after perinatal loss remains clinically unsupported in Muslim-majority contexts. It needs to be this simple. The Methods section is adequate but needs clarification. The IPA description is overly theoretical and could be condensed. Recruitment by the first author may generate power or relational bias; this needs reflexive acknowledgement. The interview question (“Please articulate your views…”) is too broad for IPA; you should clarify how probing occurred and how phenomenological depth was achieved. Ethical considerations need further clarification. Recruitment by a Muslim clinician interviewing bereaved mothers creates dual-role and dependency risks. The distress protocol should be more explicitly described. The discussion requires clearer integration of the literature, theory, and findings. Some parts repeat results rather than analyse them. A stronger Discussion would: 1) situate findings alongside cross-cultural grief literature; 2) show how Islamic perspectives mediated mothers’ decision-making; and 3) address clinical implications more analytically (e.g., culturally competent lactation counselling, protocols for milk banks, training for religious leaders). See comments on English below. Overall, I think this is a good subject and a significant contribution, but the article itself needs a lot more work. I recommend major revisions. Comments on the Quality of English LanguageEnglish-language expression also needs improvement. The manuscript is readable but contains: 1) numerous grammatical repetitions; 2) inconsistent tense; 3) overly long paragraphs; and 4) excessive quotation that could be synthesised more effectively. An extensive edit is recommended to make this more readable and less dense. Note that several references appear inconsistently formatted or incomplete. Also, consider reducing Qur’anic and hadith citations to the most relevant ones; currently, they interrupt the academic flow. Tables/figures may help condense the long thematic content. Remove repeated testimonies across multiple subsections. Also, don't forget to correct minor errors (e.g., numbering inconsistencies, stray bracketed numbers).
Author Response
Dear Authors,
Your manuscript examines an understudied and clinically significant topic and makes an original and meaningful contribution by integrating bereavement studies, lactation science, cultural competence, and religious ethics. The topic is highly relevant to clinicians, policymakers, and milk bank administrators. However, the manuscript requires structural refinement, reduction of repetition, improved cohesion between empirical findings and Islamic jurisprudential analysis, and tightening of language to meet journal standards.
- Response to Reviewer 1 Comments Point-by-point response to Comments and Suggestions for Authors Comment: Quality of English Language. The English could be improved to more clearly express the research. Response: We have improved the English throughout the article.
Response: We have improved the English throughout the article.
- Significantly, the Results section is exceptionally long and highly repetitive, reading more like a narrative review than an IPA-derived thematic presentation. IPA requires idiographic depth, but also a concise, analytic presentation. Many subsections repeat the exact quotes or thematic content. Also, integrating Islamic teachings into the Results rather than the Discussion risks conflating empirical data with interpretive frameworks.
I would suggest that you collapse the Results into 4–5 core themes, each clearly representing participant-generated meaning, with Islamic legal material shifted to Discussion. Next, Islamic jurisprudence occupies nearly half the manuscript and overshadows the qualitative findings. Although this is a key aim of the study, the balance currently obscures the mothers’ experiences.
Response: We explained in the discussion that this important question impacts Islamic legal-ethical rulings on both mothers’ choices and healthcare practices. Based on the findings presented in the manuscript and the perspectives of the religious scholars interviewed, Islamic rulings influence decision-making at two interconnected levels: the individual choices of bereaved mothers and the clinical practices of healthcare institutions, particularly lactation services and human milk banks.
The integration of Islamic jurisprudential perspectives also illuminated how ethical, legal, and emotional dimensions interact in real-world decision-making. Mothers’ reliance on scholars to interpret ridaʿa regulations reflects how religious authority structures ethical comfort and mitigates the fear of violating marriage restrictions or moral boundaries. Our findings show that when religious rulings were framed in supportive, evidence-informed ways, mothers felt empowered; when rulings were restrictive or ambiguous, emotional distress intensified. This underscores the need for clearer, standardized, and contextually sensitive religious–clinical dialogue (Page 14, 4.1, clinical implications).
The paper will be stronger if: 1) the jurisprudential explanation is summarised more concisely; detailed fatwa debates are placed in an Appendix; and the discussion clearly synthesises how these rulings influence mothers’ choices and healthcare practice.
Response: The discussion has been rewritten to explain the significance of Islamic jurisprudence on bereaved mothers’ decision to donate milk and their resolve to do so. The jurisprudential explanation has been summarized and a more detailed explanation put in an appendix (page 14, Appendix).
- Further, the Introduction needs a sharper focus and better articulates the research gap. Much background is relevant but overly expansive. The gap should be stated more clearly: 1) few studies examine bereaved Muslim mothers; 2) no studies combine IPA with Islamic legal analysis; and 3) milk donation after perinatal loss remains clinically unsupported in Muslim-majority contexts. It needs to be this simple.
Response: Text inserted after Section 1.5 Rationale for Study (Page 3).
- The Methods section is adequate but needs clarification. The IPA description is overly theoretical and could be condensed. Recruitment by the first author may generate power or relational bias; this needs reflexive acknowledgement.
Response: Methods section was clarified, Subsection 2.4.1 (page 4)The interview question (“Please articulate your views…”) is too broad for IPA; you should clarify how probing occurred and how phenomenological depth was achieved.
Response: Text added to the manuscript (2.8.) (page 6).
- Ethical considerations need further clarification. Recruitment by a Muslim clinician interviewing bereaved mothers creates dual-role and dependency risks. The distress protocol should be more explicitly described.
Response: Text added to the manuscript. This risk has been addressed in section 2.4.1. (page 4)
The study also incorporated a clearly defined distress protocol, described in 2.9.1. Participant Readiness and Emotional Well-being at Time (pages 6-7).
- The discussion requires clearer integration of the literature, theory, and findings. Some parts repeat results rather than analyse them. A stronger Discussion would: 1) situate findings alongside cross-cultural grief literature; 2) show how Islamic perspectives mediated mothers’ decision-making; and 3) address clinical implications more analytically (e.g., culturally competent lactation counselling, protocols for milk banks, training for religious leaders). See comments on English below. Overall, I think this is a good subject and a significant contribution, but the article itself needs a lot more work. I recommend major revisions.
Response: pages 12-14
Reference list was updated reference 49-60
Comments on the Quality of English Language
English-language expression also needs improvement. The manuscript is readable but contains: 1) numerous grammatical repetitions; 2) inconsistent tense; 3) overly long paragraphs; and 4) excessive quotation that could be synthesised more effectively. An extensive edit is recommended to make this more readable and less dense. Note that several references appear inconsistently formatted or incomplete. Also, consider reducing Qur’anic and hadith citations to the most relevant ones; currently, they interrupt the academic flow. Tables/figures may help condense the long thematic content. Remove repeated testimonies across multiple subsections. Also, don't forget to correct minor errors (e.g., numbering inconsistencies, stray bracketed numbers).
Response: We have improved the English throughout the article.
Editorial Refinements to Enhance Readability and Analytical Clarity
In response to reviewer feedback, the manuscript underwent extensive editorial revision to enhance clarity, coherence, and analytical focus. Redundant participant quotations were reduced, overly long paragraphs were condensed, and thematic content was reorganized for improved readability. Qur’anic and hadith citations were limited to those most relevant to the analytic argument to maintain academic flow. The reference list was standardized, and numbering inconsistencies and stray markers were corrected. Several thematic subsections were further supported with tables to summarize key findings concisely. These revisions strengthen the interpretative rigor and accessibility of the manuscript while maintaining the depth required for IPA-based qualitative research.
Round 2
Reviewer 2 Report (New Reviewer)
Comments and Suggestions for Authors
The manuscript has improved substantially and is now close to publication-ready, but some minor changes are still needed. The restructuring of the Results, the sharper articulation of the research gap, the clarification of the ethical procedures, and the refined Discussion significantly strengthen the paper. Thank you. The integration of IPA findings with Islamic legal-ethical analysis is now more coherent, and the manuscript provides an original and meaningful contribution to bereavement care, lactation counselling, and culturally competent clinical practice.
A few areas still benefit from minor revision:
- Ethics approval numbering:
Two different ethics approval codes appear in different sections of the manuscript. These should be reconciled or clearly explained. -
Results section tightening:
The restructured Results are much clearer, but some duplication remains (e.g., repeated bridging statements and overlapping quotations). Condensing these will further enhance clarity while preserving idiographic depth. -
Language and flow:
The English is generally understandable, but a light polish would help with sentence length, overuse of transitional phrasing, and a few grammatical inconsistencies. -
Minor formatting issues:
Some DOIs, spacing, and punctuation in the reference list need standardisation. This is critical in the copy stage.
Overall, the manuscript now presents a strong, coherent, and more significant contribution. Only minor refinements are required before publication.
Author Response
The manuscript has improved substantially and is now close to publication-ready, but some minor changes are still needed. The restructuring of the Results, the sharper articulation of the research gap, the clarification of the ethical procedures, and the refined Discussion significantly strengthen the paper. Thank you. The integration of IPA findings with Islamic legal-ethical analysis is now more coherent, and the manuscript provides an original and meaningful contribution to bereavement care, lactation counselling, and culturally competent clinical practice.
A few areas still benefit from minor revision:
- Ethics approval numbering:.
Reviewer’s Comment:
Ethics approval numbering: Two different ethics approval codes appear in different sections of the manuscript. These should be reconciled or clearly explained.
Response:
Thank you for this important comment. We would like to clarify that this study has one single valid ethics approval, which includes both an official approval code and an approval date. The correct approval was granted by the Institutional Review Board of Ramat Gan Academic College under approval code #2023-1010, with the approval date of 20 July 2023. The appearance of a different approval number in another section of the manuscript was an unintentional typographical error. This has now been corrected throughout the manuscript to ensure full consistency.
An adjustment was also made to the ethics number in 2.7. Ethical Considerations: approval number 2023-1010-MTA.
- Results section tightening:
The restructured Results are much clearer, but some duplication remains (e.g., repeated bridging statements and overlapping quotations). Condensing these will further enhance clarity while preserving idiographic depth.
Response to Comment :
Thank you for this helpful observation. We have carefully reviewed the entire Results section and addressed all points raised. Repetitive bridging statements and overlapping quotations have been removed, and the remaining text has been consolidated to improve clarity while maintaining the idiographic depth required by IPA methodology. The English throughout the section has also been refined for accuracy and flow. These revisions significantly streamline the narrative and ensure that each theme is presented succinctly and without duplication. Please see the revised Results section for these changes.
Language and flow:
- The English is generally understandable, but a light polish would help with sentence length, overuse of transitional phrasing, and a few grammatical inconsistencies.
Response:
Thank you for this helpful suggestion. We have carefully revised the manuscript and performed a thorough language and style edit throughout the entire text, focusing on improving sentence flow, reducing overuse of transitional phrases, and correcting grammatical inconsistencies. These changes were made across all sections of the manuscript to enhance overall clarity and readability.
Minor formatting issues:
- Some DOIs, spacing, and punctuation in the reference list need standardisation. This is critical in the copy stage.
Response to Comment:
We appreciate the reviewer’s careful attention to reference formatting. All DOIs, spacing, punctuation, and citation details have now been thoroughly reviewed and fully standardized. The entire reference list has been completely rewritten to conform strictly to the MDPI journal guidelines. These corrections ensure consistency and accuracy at the copy-editing stage. Please see the revised References section for these changes.
We have also added references 61-62 in the reference list and the paper (page 12 and page 13).
Overall, the manuscript now presents a strong, coherent, and more significant contribution. Only minor refinements are required before publication.
Response to Comment:
Minor refinements have been performed.
This manuscript is a resubmission of an earlier submission. The following is a list of the peer review reports and author responses from that submission.
Round 1
Reviewer 1 Report
Comments and Suggestions for Authors
This article addresses an important and sensitive topic: breast milk donation after perinatal loss among Muslim women, analyzed in light of Islamic ethical perspectives. The study has originality and potential significance, but the current version requires major revision before it can be considered for publication.
Key points that need improvement:
The introduction and methodology are overly detailed and redundant, despite the presentation of the sample needs to be revised because the Methods section does not provide sufficient detail about the sample.
Please specify inclusion and exclusion criteria, key socio-demographic characteristics (e.g., age, parity, education, socioeconomic status), the time from the loss at the moment of the interview, the wellbeing and the mental health of women at the moment of the interview and how saturation was determined. These details are necessary to assess the robustness and transferability of the findings.
Moreover, the results and discussion sections are compressed. Please shorten background and methods and expand the analytic depth of the results.
In the results section is really important to highlight your findings and to separate clearly between empirical findings and their interpretation; results and discussion are currently blended.
In addiction, the study strongly emphasizes positive experiences and supportive religious perspectives, but omits ambivalent or negative cases. Please acknowledge that not all mothers may find donation healing, and that some Islamic scholars remain critical of milk banks.
Consider integrating variables underexplored in your data: differences between primiparous and multiparous women, the role of fathers and extended family in decision-making.
Despite the availability of data, consider the embodied experience of lactation and milk donation after loss, in order to assess the general weelbeing in a salutogenic approach, both for the donation experience and for the subsequent pregnancies.
The article successfully integrates medical and religious perspectives, but sometimes leans toward advocacy rather than critical analysis. Please strengthen the psychosocial and sociocultural discussion (e.g., potential risks, family dynamics, cultural stigma).
The manuscript discusses riḍāʿa (milk-kinship) as understood in Islamic jurisprudence, which is valuable. However, it would strengthen the paper to explore more fully the custom of milk siblingship in practice—how “milk brothers/sisters” are recognized in different communities, how this affects the acceptance of donor milk, and whether there are local or cultural variations. Including data or literature on how people understand or experience the implications of being “milk siblings” (for instance in terms of marriage prohibitions, social identity, or familial obligations) would add depth and nuance to the religious-ethical analysis.
The heavy use of pathos (e.g., “love letter,” “ray of light”) is powerful but risks undermining the academic tone in absence of appropriate discussion around mourning process and the narrative of bereaved mothers after perinatal loss and around the entire mothers' perception of milk donation experience.
English language is understandable but could be polished for consistency and clarity.
Please check carefully the "supplementary materials"'s link: it seems not related to your research.
In conclusion, the study addresses a valuable and original question, but its presentation is currently unbalanced and overly selective. Major revision is needed to strengthen structural clarity, analytic rigor, and engagement with alternative perspectives.
Author Response
13/10/25
MDPI Healthcare Editorial Office
Dear Editor,
Thank you for the opportunity to resubmit a revised version of our manuscript. We have carefully revised the paper in accordance with the reviewers’ comments and suggestions. In addition, the manuscript has been re-edited for clarity and accuracy in English. We also revisited the interview transcripts to verify whether any additional material or data required inclusion for completeness and clarification. Below, we provide the reviewers’ comments followed by our responses, which are presented in bold. In the revised version of the manuscript, all corrections and modifications are highlighted in red.
Sincerely yours,
Mahdi Tarabeih, Mohammad Sabbah, Orsan Yahya and Khaled Awawdi
|
Response to Reviewer 1 Comments |
|
Point-by-point response to Comments and Suggestions for Authors |
|
Comment: Quality of English Language. The English could be improved to more clearly express the research. |
|
Response: We have improved the English throughout the article. |
|
Comments: This article addresses an important and sensitive topic: breast milk donation after perinatal loss among Muslim women, analyzed in light of Islamic ethical perspectives. The study has originality and potential significance, but the current version requires major revision before it can be considered for publication. Key points that need improvement. The introduction and methodology are overly detailed and redundant, despite the presentation of the sample needs to be revised because the Methods section does not provide sufficient detail about the sample. |
|
Response: We have shortened the Introduction significantly. To strengthen transparency, we revised the Participants subsection in Methods to specify inclusion and exclusion criteria more clearly. We also shortened the Methodology description. A new “Sample Characteristics” subsection was added to section 2.1 Participants (p. 3, lines 124-138) We revised Research Methodology section to include 2.5 Research design (p. 4, lines 160-171). |
|
Comment: Please specify inclusion and exclusion criteria, key socio-demographic characteristics (e.g., age, parity, education, socioeconomic status), the time from the loss at the moment of the interview, the wellbeing and the mental health of women at the moment of the interview and how saturation was determined. These details are necessary to assess the robustness and transferability of the findings. |
|
Response: We thank the reviewer for this important comment. In the revised manuscript, we expanded the Research Methodology section to provide more detailed information about the study sample, including inclusion/exclusion criteria, socio-demographics, timing of interviews, participants’ well-being, and saturation. We added 2.2 Inclusion and Exclusion Criteria: (e.g., age, parity, education, socioeconomic status), (pp. 3-4, lines 141-146) and added to section 2.9.1 with information on the time from the loss to the interview, the wellbeing and the mental health of women at the time of the interview, (p. 5, lines 216-226) and in section 2.4 how saturation was determined p. 4, lines 155-158).
|
|
Comment: Moreover, the results and discussion sections are compressed. Please shorten background and methods and expand the analytic depth of the results. |
|
1. Response: Separation of Results and Discussion: We revised the structure to ensure that the Results section presents only the empirical findings, based strictly on participants’ testimonies and inductively derived themes . A clarification paragraph was added at the end of the Results section (p. 16, lines 775-785), explicitly underscoring that no interpretative or theoretical analysis is introduced there and that all analytic meaning-making is reserved for the Discussion. The Discussion section was correspondingly refined to contain only interpretative analysis, theoretical contextualization, and clinical implications.
All these revisions are marked in red throughout the manuscript for clarity. We believe these changes address the reviewer’s concerns and strengthen the manuscript by improving structural clarity, analytic rigor, and readability. |
|
Comment: In the results section is really important to highlight your findings and to separate clearly between empirical findings and their interpretation; results and discussion are currently blended. |
|
Response: To address this, we added a paragraph at end of Results section, (p. 16, lines 775-785)
|
|
Comments: In addition, the study strongly emphasizes positive experiences and supportive religious perspectives, but omits ambivalent or negative cases. Please acknowledge that not all mothers may find donation healing, and that some Islamic scholars remain critical of milk banks. |
|
Response: We thank the reviewer for this insightful comment. We recognize the importance of presenting a balanced perspective that acknowledges not all mothers find breast milk donation to be a healing or positive experience, and that within Islamic scholarship, some critical voices remain regarding milk banking practices.To address this, we revised the Discussion section to include the following acknowledgment (added as next to last paragraph in section 4.1 , p. 17, lines 819-827) |
|
Comment: Consider integrating variables underexplored in your data: differences between primiparous and multiparous women, the role of fathers and extended family in decision-making. |
|
Response: We thank the reviewer for this thoughtful suggestion. We agree that these variables—primiparous vs. multiparous status, and the role of fathers and extended family in decision-making—are important dimensions in bereavement and milk donation experiences. While our primary analysis focused on maternal narratives and theological perspectives, we acknowledge that these additional variables enrich the contextual understanding. To address this, we have added the following paragraph in the Discussion section. (p. 17, lines 828-838). |
|
Comment: Despite the availability of data, consider the embodied experience of lactation and milk donation after loss, in order to assess the general well-being in a salutogenic approach, both for the donation experience and for the subsequent pregnancies |
|
Response: In response to the reviewer’s suggestion, we revised both the Results and Discussion sections to explicitly address the embodied experience of lactation and milk donation after perinatal loss. We also incorporated a salutogenic perspective to frame how such embodied practices influenced well-being, coping, and maternal adaptation across subsequent pregnancies. |
|
Comment: In conclusion, the study addresses a valuable and original question, but its presentation is currently unbalanced and overly selective. Major revision is needed to strengthen structural clarity, analytic rigor, and engagement with alternative perspectives. |
|
Response: We thank the reviewer for this constructive feedback. In response, we carefully revised the manuscript to improve structural clarity, strengthen analytic rigor, and incorporate engagement with alternative perspectives. All corrections and revisions have been made and are clearly marked in red throughout the article. |
Reviewer 2 Report
Comments and Suggestions for Authors
I believe the manuscript lacks of scientific approach to analyze the situation.
some sentences are not theologically correct as well.
for example:
its not one of the findings of your study. its already well-known fact.
Islamic religious scholars unanimously affirmed that milk donation is permissible 22
when milk kinship( rida’a regulations are observed, reinforcing its cultural and reli- 23
gious legitimacy.
the following result not easy to claim.
Breast milk donation after perinatal loss served as a meaningful healing ritual for 19
bereaved Muslim mothers, helping them cope with grief and maintain a symbolic 20
bond with their lost child.
since its not theology oriented journal please do not use words such as spiritually.
ealthcare providers should integrate structured counseling that includes milk do- 26
nation options to support grieving mothers emotionally and spiritually
the sample size is too small. please considering including more participants.
I believe the topic of your manuscript is not directly related to health. you may consider other journals related to religion.
Author Response
13/10/25
MDPI Healthcare Editorial Office
Dear Editor,
Thank you for the opportunity to resubmit a revised version of our manuscript. We have carefully revised the paper in accordance with the reviewers’ comments and suggestions. In addition, the manuscript has been re-edited for clarity and accuracy in English. We also revisited the interview transcripts to verify whether any additional material or data required inclusion for completeness and clarification. Below, we provide the reviewers’ comments followed by our responses, which are presented in bold. In the revised version of the manuscript, all corrections and modifications are highlighted in red.
Sincerely yours,
Mahdi Tarabeih, Mohammad Sabbah, Orsan Yahya and Khaled Awawdi
|
Response to Reviewer 2 Comments |
|
Comment: (he English could be improved to more clearly express the research. |
|
Response: We have improved the English throughout the article. |
|
Comment 1: I believe the manuscript lacks of scientific approach to analyze the situation. Some sentences are not theologically correct as well.for example: its not one of the findings of your study. its already well-known fact. Islamic religious scholars unanimously affirmed that milk donation is permissible when milk kinship (rida’a regulations are observed, reinforcing its cultural and religious legitimacy. the following result not easy to claim. |
|
Response: See Highlights. We thank the reviewer for their careful reading of our manuscript and for highlighting areas requiring clarification and correction. In revising the manuscript, we sought to ensure that it reflects a scientifically rigorous approach, avoids theological ambiguities, and clearly distinguishes between empirical findings and established scholarly knowledge. |
|
Comment: “I believe the manuscript lacks scientific approach to analyze the situation.” |
|
Response: To strengthen the scientific rigor of the study, we expanded the description of our methodological framework. In the Methods section, we added subsection 2.5 Research Design, p. 4, lines 16-171. |
|
Comment: Breast milk donation after perinatal loss served as a meaningful healing ritual for bereaved Muslim mothers, helping them cope with grief and maintain a symbolic 20 |
|
Response: We thank the reviewer for this valuable suggestion. In accordance with the comment, we carefully reviewed the entire manuscript and removed all instances of the term “spiritually” in the Highlights, Introduction, and Conclusion sections. To ensure greater scientific accuracy and to avoid theological connotations, we replaced it with the terms “emotionally,” “morally,” or “culturally,” depending on the context in which the expression appeared. These revisions enhance the clarity of the manuscript and align it more closely with the intended academic framework. |
|
Comment: The sample size is too small. please considering including more participants. |
|
Response: We acknowledge that the number of participants is relatively small; however, this is consistent with the Interpretative Phenomenological Analysis (IPA) methodology, which emphasizes depth of exploration and rich qualitative data over breadth. In qualitative bereavement research, particularly in sensitive contexts such as perinatal loss and religious-ethical considerations, smaller sample sizes are common and methodologically justified to ensure participants can share in-depth experiences without dilution of meaning. We also reached data saturation, as recurring themes and shared perceptions were observed across the interviews, suggesting that the number of participants was sufficient to capture the central lived experiences of this unique population. We agree that expanding the sample in future studies would strengthen generalizability. Therefore, we have clearly acknowledged this point under the Limitations section and recommended larger-scale, cross-cultural studies for subsequent research for enhancing generalizability. Furthermore, although the number of participants was limited, the manuscript includes a large number of direct quotations from mothers and religious scholars. This not only strengthens the trustworthiness of the findings but also ensures that participants’ voices are authentically represented. The repetition of certain quotes or themes across different interviews underscores the consistency and reliability of the shared experiences, which is a core indicator of rigor in qualitative research. |
|
Comment: I believe the topic of your manuscript is not directly related to health. you may consider other journals related to religion. |
|
Response: We thank the reviewer for this concern, yet we respectfully emphasize that the manuscript is indeed firmly situated in healthcare research, while also addressing cultural and religious contexts. This article explicitly addresses a health-related issue: bereavement care for mothers after perinatal loss. Post-loss lactation is a recognized medical and emotional challenge, and we explore how milk donation can function as therapy, easing grief while supporting infants dependent on donor milk in NICUs. These elements clearly place the work within healthcare research. At the same time, the study integrates religious-ethical perspectives. In Islamic contexts, milk donation decisions are inseparable from rulings on milk kinship (rida’a). For healthcare providers working with Muslim families, awareness of these perspectives is essential for culturally sensitive counseling and alignment of medical practices with patient values. To our knowledge, this is the first study to systematically examine breast milk donation after perinatal loss through both the voices of bereaved Muslim mothers and Islamic scholars, generating evidence-based recommendations for clinical care. By bridging health sciences with Islamic bioethics, the article fills an urgent gap in the literature of healthcare and religion in practice.For these reasons, we believe the manuscript fits well within Healthcare’s mission to address health holistically—integrating medical, psychological, ethical, and socio-cultural dimensions. |